# The Effect of Nurse Staffing on Patient Outcomes in Acute Care Hospitals in Korea

**DOI:** 10.3390/ijerph192315566

**Published:** 2022-11-23

**Authors:** Hyo-Jeong Yoon

**Affiliations:** Department of Nursing, Yeungnam University College, Daegu 42415, Republic of Korea; hjyoon@ync.ac.kr

**Keywords:** nurse staffing, acute care hospitals, length of stay, mortality, early readmission

## Abstract

Nurse staffing is an important factor influencing patient health outcomes. This study aimed to analyze the effects of nurse staffing on patient health outcomes, such as length of stay, mortality within 30 days of hospitalization, and readmission within 7 days of discharge, in acute care hospitals in Korea. Data from the first quarter of 2018 were collected using public and inpatient sample data from the Health Insurance Review and Assessment Service. The data of 46,196 patients admitted to 536 general wards of acute care hospitals were analyzed. A multilevel logistic analysis was performed for the patients’ mortality and early readmission, and a multilevel zero-truncated negative binomial analysis was performed for the length of stay. The average length of stay in acute care hospitals was 6.54 ± 6.03 days, the mortality rate was 1.1%, and the early readmission rate was 7.1%. As the nurse staffing level increased, the length of stay and number of early readmissions were likely to decrease. It can be concluded that interventions to improve nurse staffing are required; for example, a policy that compels medical institutions to comply with Korea’s medical law standards should be implemented. Additionally, continuous research and interventions are needed to establish an appropriate nurse staffing level according to patient severity.

## 1. Introduction

Inadequate nursing care is a major factor endangering patient health. According to Donabedian’s structure–process–results theory [1], nursing care is influenced by the nurse’s staffing as a structure; consequently, nurse staffing affects patient health. Empirically, improved nurse staffing has reduced patients’ adverse outcomes by preventing missed nursing care, because of factors such as adequate patient surveillance [2]. Despite continuous monitoring for early detection of exacerbations in general ward patients without improving nurse staffing, insufficient evidence of reduced patient mortality has been observed [3]. In other words, nurse staffing should be improved to reduce adverse patient outcomes. Higher nurse staffing has been associated with shorter hospital stays [4,5], fewer readmissions after discharge [6,7], and reduced mortality [8,9].

In Korea, several studies have been conducted on nurse staffing and patient health outcomes such as length of stay, readmission, and mortality. In the case of length of stay, the average length of stay was 7 days in a survey conducted in 58 acute care hospitals. A decrease in nurse staffing led to an increased length of stay; this was found to be statistically significant [4]. In addition, in an intervention study conducted in one surgical ward, one nurse was added and the patients’ hospital stay was reduced [10]. Moreover, as a result of a study using the national health insurance claim data on readmission within 30 days of discharge, the number of nurses at a medical institution increased and the readmission of its patients decreased [11]. Early readmissions within seven days rather than 30 days after discharge are preventable [12]; hence, a study on the relationship between nurse staffing and early readmission is needed. Lastly, with regard to mortality, a statistically significant correlation was reported between nurse staffing and patient mortality in various studies such as patients admitted to general wards or intensive care units or patients with stroke or cardiovascular disease [13,14,15,16]. However, most of the researches have been limited to medical institutions, such as general or tertiary hospitals, or to patients with specific diseases. 

Since 1999, Korea has implemented a policy of differential nursing management fees based on the nurse staffing level, which is determined by the average ratio of the number of beds to the number of nurses in the general ward of acute care hospitals. In other words, all patients admitted to the general ward should be included, not just patients with specific diseases. Furthermore, it is necessary to understand the health outcomes of patients in all medical institutions in acute care hospitals such as tertiary hospitals, general hospitals, and hospitals. The hospitalized patient sample data provided by the Health Insurance Review and Assessment Service contain information on all health insurance medical expenses incurred by patients over the course of the year, which allows us to study patient readmission. In addition, through the admission fee charged by the medical institution where the patient was admitted, it is possible to know the status of nurse staffing in the medical institution. Therefore, this study aimed to analyze the status of the length of stay, readmission, and mortality, which are the outcomes of patients admitted to all acute care hospitals, and to investigate their relationship with nurse staffing.

## 2. Materials and Methods

### 2.1. Design and Database

This was a retrospective cross-sectional correlational study. The target medical institutions of this study were tertiary hospitals, general hospitals, and hospitals, and the patients were adults who were admitted to general wards. Studying hospital readmission requires country-level data because patients may move to other hospitals in other regions. Since the national health insurance service in Korea is a single insurer mandatory for all citizens, it collects data on the insurance used by all medical institutions of patients in order that readmissions can be confirmed. The Health Insurance Review and Assessment Service de-identified and provided an inpatient sample of national health insurance claims data of 10% of inpatients for academic purposes [17]. Since April 2018, the basis for calculating nursing management fees according to nurse staffing levels has changed from the number of beds to the number of patients, except for tertiary hospitals. Therefore, to include all acute care hospitals, this study only used data corresponding to the first quarter of 2018. These data were based on the inpatient sample data of the Health Insurance Review and Assessment Service (HIRA-NIS-2018-0037), and the research results were not related to the Health Insurance Review and Assessment Service or the Ministry of Health and Welfare.

### 2.2. Data Collection

The participants were adult patients in general wards and data were collected through seven steps. First, in January, February, and March 2018, participants who requested admission to general wards at all acute care hospitals were selected. Second, patients aged less than 20 years were excluded from the study. Third, patients whose departments were pediatrics, obstetrics and gynecology, or psychiatry were excluded. The first hospitalization of each patient was designated as an index episode. Fourth, patients who died within 30 days of hospitalization in December 2017 were excluded from the study. For this purpose, the wash-out period was designated as January 2018 and January inpatients were excluded. Fifth, patients who were discharged on the same day as admission and those who stayed for more than 30 days were excluded. Sixth, patients with missing data on major variables were excluded. Finally, medical institutions with fewer than 30 patients were excluded.

### 2.3. Variables

#### 2.3.1. Hospital Level

The characteristics of the hospital chosen included the hospital type, region, and nurse staffing level in the general ward. Hospitals were categorized into tertiary hospitals, general hospitals, and hospitals. Hospital regions were divided into capital and non-capital regions. The capital region includes Seoul, Incheon, and Gyeonggi-do, while the non-capital region includes the rest of the country. Nurse staffing level in a general ward was divided into levels 1 through 7 according to the bed-to-nurse ratio. In the case of general hospitals and hospitals, nurse staffing level 1 is 2.5:1 or less, level 2 is between 2.5:1 and 3.0:1, level 3 is between 3.0:1 and 3.5:1, level 4 is between 3.5:1 and 4.0:1, level 5 is between 4.0:1 and 4.5:1, level 6 is between 4.5:1 and 6.0:1, and level 7 is 6.0:1 or more. In tertiary hospitals, nurse staffing level 1 is 2.0:1 or less, which is one level higher than that in general hospitals and hospitals. To apply the same standard for the bed-to-nurse ratio, the nurse staffing level in the tertiary hospital was lowered by one level. That is, level 1 of the tertiary hospital was changed to level 0.

#### 2.3.2. Patient Level

The patients’ independent variables included age, sex, health insurance type, route of admission, intensive care unit admission, and comorbidities or complications. Health insurance types were divided into medical aid and health insurance, and medical aid had a low social economy. The route of admission was divided into emergency room admissions, for which emergency medical management fees were charged, and outpatient admissions. Admission to the intensive care unit was also classified according to whether the intensive care unit admission fee was charged. Complications or comorbidities were measured using the Korean Diagnosis-Related Group (KDRG) codes and classified as none (0), mild (1), moderate (2), or severe (3).

The dependent variables were as follows: The length of stay was from the 1st to the 30th day, except when the patient was discharged on the day of admission. A 30-day mortality refers to death within 30 days of the date of admission; a 7-day readmission refers to readmission to an acute care hospital within 7 days of the date of discharge.

### 2.4. Data Analysis

The characteristics of the patients and medical institutions and dependent variables were analyzed using descriptive statistics (frequency, percentage, average, and standard deviation). The differences in the length of stay according to the characteristics of patients and medical institutions were analyzed using a t-test and ANOVA. The difference between the mortality and readmission was analyzed using the chi-square test. The relationship between nurse staffing and patient outcomes was analyzed using a multilevel model. Since patients were hierarchically nested in a specific medical institution, a multilevel model was applied to correct the violation of the patient’s independence assumption that occurred while sharing the characteristics of the medical institution. The length of stay was measured using a multilevel zero-truncated negative binomial model and mortality (“yes” = 1, “no” = 0) and readmission (“yes” = 1, “no” = 0) were analyzed through multilevel logistic regression. The analysis was conducted using SAS software (SAS Institute, Cary, NC, USA).

### 2.5. Ethical Considerations

This study analyzed data after receiving approval from the institutional review board of the first author university (YNC IRB-202006-07). 

## 3. Results

### 3.1. Hospital Characteristics

Table 1 summarizes the general characteristics of the acute care hospitals included in this study. The nurse staffing level of 536 acute care hospitals is distributed from level 0 to level 7, with level 7 being the most common at 27.8%, followed by level 6 at 16.0%. By hospital type, 42 tertiary hospitals, 253 general hospitals, and 241 hospitals were included, and most acute care hospitals were located in non-metropolitan areas (63.2%).

Of the 46,196 patients admitted to acute care hospitals, the highest number was 10,627 (23.0%) at level 1, followed by 7425 (16.1%) at level 7. Most patients were admitted to general hospitals (*n* = 22,566, 48.8%) and acute care hospitals located in non-metropolitan areas (*n* = 26,462, 57.3%).

### 3.2. Length of Stay, Mortality, and Readmissions According to the Characteristics of Medical Institutions

Table 2 shows the results of the univariate analysis of the length of stay, 30-day mortality, and 7-day readmission according to hospital characteristics. There was a statistically significant difference in lengths of stay depending on the nurse staffing level, type, and location of the hospital. The average length of stay was 6.54 ± 6.03 days. Lengths of stay for level 0 (5.00 ± 4.59), level 1 (6.23 ± 5.57), and level 5 (6.38 ± 6.17) institutions were shorter than the average. Regarding the type of hospital, tertiary hospitals (5.84 ± 5.27) had the shortest length of stay and general hospitals (6.87 ± 6.32) had the longest. Hospitals located in the capital area (5.88 ± 5.67) reported shorter stays than those in non-capital areas (7.03 ± 6.24).

The 30-day mortality rate showed a statistically significant difference occurring from differences in nurse staffing and hospital type. The overall 30-mortality rate was 1.1%. The nurse staffing level 5 had the lowest (0.5%), whereas level 3 had the highest mortality rate (1.5%). According to the type of hospital, hospitals had the lowest mortality rate at 0.5% and general hospitals had the highest at 1.3%.

As a result of analyzing 45,805 patients, excluding 391 who died within 7 days of discharge, there were statistical differences depending on the nurse staffing level and the location of the hospital. The number of patients 7-day-readmitted to acute care hospitals was 3189 (7.1%). The nurse staffing level was lowest in level 5 (6.3%) and highest in level 7 (8.1%). Hospitals located in capital areas (6.0%) had fewer readmissions than those in non-capital areas (7.9%).

### 3.3. Relationship between Nurse Staffing and Length of Stay, Mortality, and Readmissions

Table 3 presents the results of the multilevel analysis of the relationship between hospital characteristics, length of stay, death, and readmission. There was a statistically significant difference in the length of stay depending on the nurse staffing level and location of the hospital. Assuming that all characteristics are the same except for nurse staffing level, the length of stay at level 0 was 76.7% of that at level 7, which had the lowest nurse staffing level. In other words, the length of stay of patients admitted to level 0 was 23.3% shorter than that of patients admitted to level 7. Similarly, the length of stay decreased by 17.6% for level 1, 16.2% for level 2, 13.5% for level 3, 10.0% for level 4, and 12.5% for level 5, in comparison with level 7. In addition, the length of stay was reduced by 15.8% in the capital area compared with patients admitted to a medical institution located in non-capital areas. There were no statistically significant differences in hospital characteristics for 30-day mortalities. 

Nurse staffing level, type, and location of the hospital were statistically significant variables for readmission to an acute care hospital within 7 days of discharge. Assuming that all characteristics were the same except for the nurse staffing level, patients from level 1 and level 3 institutions were less likely to be readmitted than patients from level 7 institutions. The likelihood of 7-day readmission was low for general hospitals and medical institutions located in capital areas.

## 4. Discussion

This study investigated the length of stay, 30-day mortality, and 7-day readmission of adult patients in the general wards of acute care hospitals nationwide in the first quarter of 2018. In previous studies, participants were limited to those admitted to some tertiary or general hospitals, but in this study, all acute care hospitals were analyzed using health insurance data. The readmission by the medical institution was also confirmed, including all patients readmitted to acute hospitals in other regions after discharge.

We found that hospital characteristics had a statistically significant relationship with the length of stay, 30-day mortality, and 7-day readmission. Among the characteristics of medical institutions, the nurse staffing level was an important one. The higher the nurse staffing level, the shorter the length of stay and the lower the likelihood of readmission. Increased nurse staffing decreases the chances of adverse events that result from a failure to monitor and enables rapid treatment [2]. This is why some countries have already implemented mandatory patient-to-nurse ratio policies [18]. When the level of nurse staffing was raised through legislation, the length of stay of patients was reduced, the readmission rate was lowered, and the 30-day mortality rate was reduced [19].

In this study, we found that nurse staffing was associated with the length of stay and readmission rates. Lower nurse staffing increases the likelihood of adverse events occurring with the patients [20], whereas higher nurse staffing prevents these adverse events as there are fewer chances of delay or omission of care [2]. Previous studies found that the higher the nurse staffing, the shorter the length of stay [5]; this was also the case in this study. Moreover, when a patient’s health deteriorates, such as an instability in vital signs, patients who receive insufficient nursing care are more likely to be transferred to another acute care hospital or have a short-term readmission after discharge [21]. The lower the nurse staffing in hospitals where patients are not provided with sufficient care, the higher the likelihood of early readmission after discharge [7].

Contrary to the results of previous studies showing that the higher the nurse staffing, the lower the patients’ mortality rate [8,9], this association was not statistically significant in this study. The reason for this discrepancy is likely to be the high short-term readmission rate. In a recent study conducted in Queensland, the mortality rate within 30 days of hospitalization was 1.1–1.6%, and the readmission rate within 7 days of discharge was 3.2% [19]. This study had a mortality rate of 1.0%, slightly lower than that reported in previous studies, but the readmission rate was 7.1%, which is more than twice that of the previous study. A possible explanation for this is that the patient’s health deteriorates, which leads to their death in the hospital where they were first admitted rather than when they are transferred to another acute care hospital. The Korean medical system classifies tertiary, general, and other acute care hospitals according to the severity of the patients, even within acute care hospitals, and encourages admission to hospitals tailored to the patients’ severity [22]. However, even if the type of medical institution is the same, there is a difference in early readmission to medical institutions dependent on nurse staffing. Patients who undergo an early readmission have a higher risk of death than other patients [23]; therefore, management of nurse staffing is necessary.

Appropriate nurse staffing has a positive impact on patient outcomes if care is provided to patients without failure. Some countries have implemented, through legislation, a compulsory minimum nurse-to-patient ratio to provide sufficient care to patients [18]. Nurses should not attend to more patients than the stipulated nurse-to-patient ratio. In Korea, there is a difference in the application of medical law based on the average number of hospitalized patients per day divided by 2.5 for registered nurses working in hospitals. In contrast to the aforementioned nurse-to-patient ratio legislation, the average number of hospitalized patients per day in Korea is based on the number of beds, not the number of patients. Further, the number of registered nurses is calculated based on the annual average, not the number of shifts. Since the medical act is not a compulsory provision, nurse staffing is being improved through a financial incentive called differential nursing management fees. These fees are based on the nurse staffing levels. This differs from enacted nurse-to-patient ratio legislation because it is based on the number of beds, not the number of patients, and it is calculated as an annual average rather than the maximum number of patients per shift. According to the medical act, similar to the nurse-to-patient ratio, the number of patients per nurse by shift is 11.9. However, the estimated number of patients per nurse by shift is 16.3 for tertiary and general hospitals, and 43.6 for hospitals [24]. Compared with a nurse-to-patient ratio of 5.3 in the US, 8.6 in the UK, and 13.0 in Germany [25], Korean nurses care for the largest number of patients. In other words, the nursing hours per day (NHPPD) for one patient are low, making it difficult to provide necessary nursing care to the patient [26]. Instead of a nurse, the patient’s family or caregiver is forced to assist with nursing care. Therefore, nurse staffing should be improved to ensure sufficient nursing care for improved patient outcomes.

This study had some limitations. First, nurse staffing could not be presented as the nurse-to-patient ratio or the number of nursing hours per patient (NHPPD) but as presented in health insurance data, in other words, the number of beds per nurse. Depending on the bed occupancy rate of a medical institution, the deviation in actual nurse staffing increases. From April 2018 (Q2 2018), the nurse staffing level has changed from bed per nurse to patient per nurse in Korea. However, some medical institutions are still applying beds per nurse. Additionally, some of the exclusions were lifted in October 2019. An increasing number of medical institutions are calculating the nurse level based on patients rather than beds [27]. Therefore, when analyzing post-2020 data, most acute care hospitals in Korea can be included, and the nurse staffing level can be applied to patients per nurse; therefore, future research is needed. Second, medical institutions with fewer than 30 episodes of hospitalization were excluded. If there is a certain bias in nurse staffing and patient outcomes in the excluded medical institutions, the results of this study may be affected. Despite these limitations, this study is meaningful in that it provides a basis for changing nurse staffing in medical institutions to improve patient outcomes.

## 5. Conclusions

This study showed that the higher the nurse staffing, the better the patient’s outcomes. The lower the nurse staffing, the longer the length of stay and the greater the increase in early readmission to acute care hospitals. These findings have implications for the Korean government. The Korean government should supervise medical institutions to ensure that they have adequate nurses for patients’ safety and health. The government should, first, enforce compliance with nurse staffing under the medical act and additionally implement policies to improve nurse staffing. It is hoped that this will enable nurses to improve patient outcomes by providing quality care to patients.

## Figures and Tables

**Table 1 ijerph-19-15566-t001:** General characteristics of medical institutions and patients.

		No. of Patients (%)	No. of Medical Institutions (%)
Nurse staffing	Level 0	3623 (7.8)	6 (1.1)
	Level 1	10,627 (23.0)	52 (9.7)
	Level 2	6665 (14.4)	64 (11.9)
	Level 3	6812 (14.7)	85 (15.9)
	Level 4	3918 (8.5)	56 (10.4)
	Level 5	2282 (4.9)	38 (7.1)
	Level 6	4844 (10.5)	86 (16.0)
	Level 7	7425 (16.1)	149 (27.8)
Type of hospital	Tertiary hospitals	12,103 (26.2)	42 (7.8)
	General hospitals	22,566 (48.8)	253 (47.2)
	Hospitals	11,527 (25.0)	241 (45.0)
Location	Capital area	19,734 (42.7)	195 (36.4)
	Non-capital area	26,462 (57.3)	341 (63.6)

**Table 2 ijerph-19-15566-t002:** Comparison of length of stay, mortality, and readmission according to characteristics.

		Length of Stay (Days) (*N* = 46,196)		30-Day Mortality (*N* = 46,196)		7-Day Readmission (*N* = 45,805)	
Overall		6.54 ± 6.03		486/46,196 (1.1)		3244/45,805 (7.1)	
Age	≥75	8.49 ± 6.76	<0.001	314/8904 (3.5)	<0.001	873/8648 (10.1)	<0.001
	65–74	7.21 ± 6.36		90/8390 (1.1)		686/8319 (8.2)	
	20–64	5.74 ± 5.51		82/28,902 (0.3)		1685/28,838 (5.8)	
Gender	Male	6.23 ± 5.80	<0.001	261/22,523 (1.2)	0.028	1599/22,318 (7.2)	0.503
	Female	6.83 ± 6.22		225/23,673 (1.0)		1645/23,487 (7.0)	
Type of insurance	Medical aid	8.95 ± 6.97	<0.001	58/3360 (1.7)	<0.001	336/3318 (10.1)	<0.001
	Health insurance	6.35 ± 5.90		428/42,836 (1.0)		2908/42,487 (6.8)	
Route of admission	Emergency room	8.05 ± 6.56	<0.001	280/10,215 (2.7)	<0.001	1106/9982 (11.1)	<0.001
	Outpatient department	6.10 ± 5.79		206/35,981 (0.6)		2138/35,823 (6.0)	
Admitted to an intensive care unit	Yes	11.71 ± 7.33	<0.001	116/2035 (5.7)	<0.001	223/1925 (11.6)	<0.001
	No	6.30 ± 5.85		370/44,161 (0.8)		3021/43,880 (6.9)	
Comorbidities or complications	Severe	9.49 ± 6.72	<0.001	173/3192 (5.4)	<0.001	381/3050 (12.5)	<0.001
	Moderate	7.89 ± 6.44		141/7722 (1.8)		723/7610 (9.5)	
	Mild	7.02 ± 6.28		70/10,579 (0.7)		791/10,527 (7.5)	
	None	5.52 ± 5.42		102/24,703 (0.4)		1349/24,618 (5.5)	
Nurse staffing	Level 0	5.00 ± 4.59	<0.001	29/3623 (0.8)	<0.001	249/3606 (6.9)	0.003
	Level 1	6.23 ± 5.57		137/10,627 (1.3)		702/10,513 (6.7)	
	Level 2	6.52 ± 6.08		101/6665 (1.5)		475/6584 (7.2)	
	Level 3	6.64 ± 6.27		67/6812 (1.0)		467/6754 (6.9)	
	Level 4	6.57 ± 6.26		35/3918 (0.9)		249/3891 (6.4)	
	Level 5	6.38 ± 6.17		12/2282 (0.5)		142/2272 (6.3)	
	Level 6	7.08 ± 6.22		47/4844 (1.0)		361/4813 (7.5)	
	Level 7	7.30 ± 6.50		58/7425 (0.8)		599/7372 (8.1)	
Type of institution	Tertiary hospitals	5.84 ± 5.27	<0.001	136/12,103 (1.1)	<0.001	840/11,996 (7.0)	0.345
	General hospitals	6.87 ± 6.32		292/22,566 (1.3)		1619/22,328 (7.3)	
	Hospitals	6.61 ± 6.12		58/11,527 (0.5)		785/11,481 (6.8)	
Location	Capital area	5.88 ± 5.67	<0.001	214/19,734 (1.1)	0.556	1168/19,564 (6.0)	<0.001
	Non-capital area	7.03 ± 6.24		272/26,462 (1.0)		2076/26,241 (7.9)	

Note. 7-day Readmission was analyzed after excluding patients who died within 7 days of discharge.

**Table 3 ijerph-19-15566-t003:** Factors related to the length of stay, mortality, and readmission.

		Length of Stay (Days) (*N* = 46,196)		30-Day Mortality (*N* = 46,196)		7-Day Readmission (*N* = 45,805)	
Overall		6.54 ± 6.03		486/46,196 (1.1)		3244/45,805 (7.1)	
Age	≥75	1.30 (1.27–1.34)	<0.001	7.16 (5.52–9.28)	<0.001	1.38 (1.26–1.52)	<0.001
	65–74	1.23 (1.20–1.27)	<0.001	2.90 (2.14–3.94)	<0.001	1.31 (1.19–1.44)	<0.001
	20–64	1.00		1.00		1.00	
Gender	Male	0.92 (0.91–0.94)	<0.001	1.43 (1.19–1.73)	<0.001	1.05 (0.97–1.13)	0.216
	Female	1.00		1.00		1.00	
Type of insurance	Medical aid	1.31 (1.27–1.36)	<0.001	1.19 (0.89–1.59)	<0.001	1.29 (1.14–1.46)	<0.001
	Health insurance	1.00		1.00		1.00	
Route of admission	Emergency room	1.23 (1.20–1.26)	<0.001	2.10 (1.71–2.58)	<0.001	1.77 (1.62–1.93)	<0.001
	Outpatient department	1.00		1.00		1.00	
Admitted to an intensive care unit	Yes	1.85 (1.77–1.94)	<0.001	3.00 (2.37–3.80)	<0.001	1.24 (1.07–1.45)	0.006
	No	1.00		1.00		1.00	
Comorbidities or complications	Severe	1.58 (1.52–1.65)	<0.001	4.79 (3.65–6.28)	<0.001	1.89 (1.66–2.16)	<0.001
	Moderate	1.38 (1.34–1.41)	<0.001	2.16 (1.65–2.83)	<0.001	1.53 (1.38–1.69)	<0.001
	Mild	1.26 (1.23–1.30)	<0.001	1.10 (0.81–1.50)	.542	1.29 (1.18–1.42)	<0.001
	None	1.00		1.00		1.00	
Nurse staffing	Level 0	0.77 (0.60–0.98)	0.035	1.02 (0.51–2.02)	0.956	1.07 (0.74–1.55)	0.707
	Level 1	0.82 (0.73–0.93)	0.002	1.08 (0.68–1.72)	0.736	0.75 (0.60–0.94)	0.011
	Level 2	0.84 (0.77–0.91)	<0.001	1.10 (0.75–1.62)	0.625	0.86 (0.73–1.02)	0.091
	Level 3	0.86 (0.80–0.93)	<0.001	0.90 (0.61–1.34)	0.602	0.83 (0.71–0.98)	0.026
	Level 4	0.90 (0.82–0.98)	0.019	1.02 (0.65–1.61)	0.932	0.84 (0.70–1.01)	0.068
	Level 5	0.88 (0.79–0.97)	0.010	0.68 (0.36–1.31)	0.247	0.81 (0.65–1.01)	0.058
	Level 6	0.95 (0.88–1.02)	0.146	1.18 (0.79–1.77)	0.426	0.90 (0.77–1.06)	0.200
	Level 7	1.00		1.00		1.00	
Type of institution	Tertiary hospitals	0.96 (0.83–1.10)	0.530	1.19 (0.71–1.97)	0.512	1.00 (0.79–1.27)	0.987
	General hospitals	1.00 (0.95–1.06)	0.967	1.17 (0.83–1.64)	0.377	0.87 (0.77–0.99)	0.027
	Hospitals	1.00		1.00		1.00	
Location	Capital area	0.84 (0.80–0.89)	<0.001	1.12 (0.90–1.40)	0.294	0.75 (0.67–0.83)	<0.001
	Non-capital area	1.00		1.00		1.00	

## Data Availability

Not applicable.

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
