# Peer review of "The Effect of Nurse Staffing on Patient Outcomes in Acute Care Hospitals in Korea"

_ijerph, 2022, doi:10.3390/ijerph192315566_

Round 1

Reviewer 1 Report

Page 8 lines 261-3 is confusing stating "However, according to reports from medical institutions, it is estimated that general hospitals have a ratio of 16.3, and hospitals 43.6 [24]."  The use of General Hospitals and Hospitals need better definition here.

The authors note a limitation of the study because Korean nursing ratios are related to hospital beds which is not consistent with other country studies that report the ratios of nursing numbers to the number of patients.  It would seem that varying occupancy rates of Korean hospitals would create havoc when trying to relate nursing numbers to patient numbers if occupancy rates varied widely.

Despite the above comment, the conclusions of the authors suggest that the problem I mentioned above, have not caused the Korean numbers to divert significantly from other country numbers.  In addition, the conclusion that calculations based upon on just hospital bed numbers as they relate to the number of nurses concerns me because hospital occupancy rates could vary widely, possibly reducing the value of this very important article.     

Author Response

Thank you for giving me the opportunity to submit a revised draft of my manuscript titled The effect of nurse staffing on patient outcomes in acute care hospitals in Korea to International Journal of Environmental Research and Public Health. I appreciate the time and effort that reviewers have dedicated to providing your valuable feedback on my manuscript. I am are grateful to the reviewers for their insightful comments on my paper. I have been able to incorporate changes to reflect most of the suggestions provided by the reviewers. I have highlighted the changes within the manuscript. 

Here is a point-by-point response to the reviewers’ comments and concerns.

Response to Reviewer 1 Comments

Point 1: Page 8 lines 261-3 is confusing stating "However, according to reports from medical institutions, it is estimated that general hospitals have a ratio of 16.3, and hospitals 43.6 [24]."  The use of General Hospitals and Hospitals need better definition here.

Response 1: I appreciate your comments. I have revised the part as follows:

According to the medical act, similar to the nurse-to-patient ratio, the number of patients per nurse by shift is 11.9. However, the estimated number of patients per nurse by shift is 16.3 for tertiary and general hospitals, and 43.6 for hospitals [24]. (page 8, lines 264-267)

Point 2: The authors note a limitation of the study because Korean nursing ratios are related to hospital beds which is not consistent with other country studies that report the ratios of nursing numbers to the number of patients. It would seem that varying occupancy rates of Korean hospitals would create havoc when trying to relate nursing numbers to patient numbers if occupancy rates varied widely.

Despite the above comment, the conclusions of the authors suggest that the problem I mentioned above, have not caused the Korean numbers to divert significantly from other country numbers. In addition, the conclusion that calculations based upon on just hospital bed numbers as they relate to the number of nurses concerns me because hospital occupancy rates could vary widely, possibly reducing the value of this very important article.

Response 2: Thank you for this pertinent comment.

Consistent with the reviewer's concern, the variation in the number of patients treated by nurses based on bed occupancy rate was among the issues in this article. From April 2018 (Q2 2018), the nurse staffing level was changed from 'bed per nurse' to 'patient per nurse'. However, medical institutions that fall under one or more of 1) specific regions, 2) specific laws and regulations, and 3) local emergency are excluded. Excluded medical institutions were applied as 'beds per nurse' as before. In other words, from the second quarter of 2018, the method of calculating the nurse staffing level in Korea has changed depending on the characteristics of medical institutions. In order to analyze all acute hospitals in Korea, this study analyzed data from the first quarter of 2018 by applying the same nurse staffing level calculation.

In addition, from October 2019, exclusion conditions 1) of specific regions have been lifted. Accordingly, an increasing number of medical institutions are calculating nurse staffing level based on the number of patients per nurse. According to a recently reported study, In the first quarter of 2020, the nurse staffing level was calculated as patients per nurse in 92.4% of tertiary and general hospitals. The above medical institutions do not need to consider the variation according to bed occupancy in nurse staffing and do not have to consider the variation according to bed occupancy in nurse assignments. Analysis of data from 2020 and beyond, covering most acute care hospitals, can confirm the relationship between nurse staffing level and patient health outcomes. I have suggested this in this article and will conduct research with the latest data as soon as possible.

I have adddressed the reviewer’s concern in the manuscript as follows:

First, nurse staffing could not be presented as the nurse-to-patient ratio or the number of nursing hours per patient (NHPPD) but as presented in health insurance data, which is number of beds per nurse. Depending on the bed occupancy rate of a medical institution, the deviation in actual nurse staffing increases. From April 2018 (Q2 2018), the nurse staffing level has changed from bed per nurse to patient per nurse in Korea. However, some medical institutions are still applying beds per nurse. Additionally, some of the exclusions were lifted in October 2019. An increasing number of medical institutions are calculating nurse level based on patients rather than beds [27]. Therefore, when analyzing post-2020 data, most acute care hospitals in Korea can be included, and the nurse staffing level can be applied to patients per nurse; therefore, future research is needed. (page 8,lines 273-283)

Reviewer 2 Report

Recommendation: Accept with Minor Revision

Dear author/s,

Thank you for the opportunity to review this manuscript. I hope you find my comments useful as you consider revising the paper. The topic is fitting with the aim and scope of the Journal. I hope this review provides some useful feedback and wish you the best of luck with the development of this paper!

Additional Questions:

1.      Originality:  Does the paper contain new and significant information adequate to justify publication?: Yes. he authors identified an interesting topic and provided significant information on the relation of The effect of nurse staffing on patient outcomes in acute care hospitals in Korea”. However, there are some issues that need clarification. More importantly, the discussion regarding the research gaps is missing!

2.      Relationship to Literature:  Does the paper demonstrate an adequate understanding of the relevant literature in the field and cite an appropriate range of literature sources?  Is any significant work ignored?: The literature review section is fine.

3.      Methodology:  Is the paper's argument built on an appropriate base of theory, concepts, or other ideas?  Has the research or equivalent intellectual work on which the paper is based been well designed?  Are the methods employed appropriate?: The paper is well structured and follows the standards.

4. Results:  Are results presented clearly and analysed appropriately?  Do the conclusions adequately tie together the other elements of the paper?: The results section is fine.

5. Implications for research, practice and/or society:  Does the paper identify clearly any implications for research, practice and/or society?  Does the paper bridge the gap between theory and practice? How can the research be used in practice (economic and commercial impact), in teaching, to influence public policy, in research (contributing to the body of knowledge)?  What is the impact upon society (influencing public attitudes, affecting quality of life)?  Are these implications consistent with the findings and conclusions of the paper?: The implications part missing.

6. Quality of Communication:  Does the paper clearly express its case, measured against the technical language of the field and the expected knowledge of the journal's readership?  Has attention been paid to the clarity of expression and readability, such as sentence structure, jargon use, acronyms, etc.: A professional review of the language is strongly suggested because several parts of the text are unclear.

Author Response

Thank you for giving me the opportunity to submit a revised draft of my manuscript titled The effect of nurse staffing on patient outcomes in acute care hospitals in Korea to International Journal of Environmental Research and Public Health. I appreciate the time and effort that reviewers have dedicated to providing your valuable feedback on my manuscript. I am are grateful to the reviewers for their insightful comments on my paper. I have been able to incorporate changes to reflect most of the suggestions provided by the reviewers. I have highlighted the changes within the manuscript. 

Here is a point-by-point response to the reviewers’ comments and concerns.

Response to Reviewer 2 Comments

Response to Reviewer 2 Comments

  1. Originality:Does the paper contain new and significant information adequate to justify publication?: Yes. he authors identified an interesting topic and provided significant information on the relation of “The effect of nurse staffing on patient outcomes in acute care hospitals in Korea”. However, there are some issues that need clarification. More importantly, the discussion regarding the research gaps is missing!

Response 1: Thank you for the comment. The suggested change has been made.

This study investigated length of stay, 30-day mortality, and 7-day readmission of adult patients in general wards of acute care hospitals nationwide in the first quarter of 2018. In previous studies, participants were limited to those admitted to some tertiary or general hospitals, but in this study, all acute care hospitals were analyzed using health insurance data. The readmission by medical institution was also confirmed, including all patients readmitted to acute hospitals in other regions after discharge. (page 7,lines 209-214)

  1. Relationship to Literature:Does the paper demonstrate an adequate understanding of the relevant literature in the field and cite an appropriate range of literature sources?  Is any significant work ignored?: The literature review section is fine.

Response 2: I thank you for your comment.

  1. Methodology:Is the paper's argument built on an appropriate base of theory, concepts, or other ideas?  Has the research or equivalent intellectual work on which the paper is based been well designed?  Are the methods employed appropriate?: The paper is well structured and follows the standards.

Response 3: I appreciate your comments.

  1. Results: Are results presented clearly and analysed appropriately? Do the conclusions adequately tie together the other elements of the paper?: The results section is fine.

Response 4: I thank the reviewer.

  1. Implications for research, practice and/or society: Does the paper identify clearly any implications for research, practice and/or society? Does the paper bridge the gap between theory and practice? How can the research be used in practice (economic and commercial impact), in teaching, to influence public policy, in research (contributing to the body of knowledge)?  What is the impact upon society (influencing public attitudes, affecting quality of life)?  Are these implications consistent with the findings and conclusions of the paper?: The implications part missing.

Response 5: Thank you for pointing this out. I have included the implication of the study.

This study showed that the higher the nurse staffing, the better the patient's outcomes. The lower the nurse staffing, the longer the length of stay and the greater the increase in early readmission to acute care hospitals. These findings have implicatin for the Korean government. The Korean government should supervise medical institutions to ensure that they have adequate nurses for patients’ safety and health. The government should ,first, enforce compliance with the nurse staffing under the medical act and additionally implement policies to improve the nurse staffing. It is hoped that this will enable nurses to improve patient outcomes by providing quality care to patients. (pages 8-9, lines 290-297)

  1. Quality of Communication: Does the paper clearly express its case, measured against the technical language of the field and the expected knowledge of the journal's readership? Has attention been paid to the clarity of expression and readability, such as sentence structure, jargon use, acronyms, etc.: A professional review of the language is strongly suggested because several parts of the text are unclear.

Response 6: Thank you for your suggestion. I used an English editing service (Editage) to address the grammar and tense issues.
